# Feature Consistent Point Cloud Registration in Building Information Modeling

**DOI:** 10.3390/s22249694

**Published:** 2022-12-10

**Authors:** Hengyu Jiang, Pongsak Lasang, Georges Nader, Zheng Wu, Takrit Tanasnitikul

**Affiliations:** 1School of Computer Science and Engineering, Nanjing University of Science and Technology, Nanjing 210000, China; 2Panasonic R&D Center Singapore, Singapore 469332, Singapore

**Keywords:** point cloud registration, building information modeling, feature consistent

## Abstract

Point Cloud Registration contributes a lot to measuring, monitoring, and simulating in building information modeling (BIM). In BIM applications, the robustness and generalization of point cloud features are particularly important due to the huge differences in sampling environments. We notice two possible factors that may lead to poor generalization, the normal ambiguity of boundaries on hard edges leading to less accuracy in transformation; and the fact that existing methods focus on spatial transformation accuracy, leaving the advantages of feature matching unaddressed. In this work, we propose a boundary-encouraging local frame reference, the PyramidFeature(PMD), consisting of point-level, line-level, and mesh-level information to extract a more generalizing and continuous point cloud feature to encourage the knowledge of boundaries to overcome the normal ambiguity. Furthermore, instead of registration guided by spatial transformation accuracy alone, we suggest another supervision to extract consistent hybrid features. A large number of experiments have demonstrated the superiority of our PyramidNet (PMDNet), especially when the training (ModelNet40) and testing (BIM) sets are very different, PMDNet still achieves very high scalability.

## 1. Introduction

Optical 3D scanning shows a growing trend in the construction industry, providing a complete and consistent building engineering database by establishing a virtual building 3D model based on digital technology. Nowadays, Building Information Modeling (BIM), as a key application of optical 3D scanning, dominates project follow-up, building monitoring, and maintenance [1,2,3,4], architecture planning [5,6,7,8], emergency simulation [9,10,11,12], and IoT [13,14,15,16]. Implementing BIM methods helps improve the efficiency and integration of constructions in all stages of their life-cycle, as well as helps provide a platform for engineering information exchange and communication [17,18,19,20].

This work is focused on point cloud registration in building information modeling. In practice, one of the key tasks to solve is how to align pre-built 3D models against the scanned points of real buildings. The aligning results help generate complete and accurate digital models, contributing largely to measuring and applications such as simultaneous localization and mapping (SLAM), 3D reconstruction [21,22,23,24], localization [25,26,27,28], and pose estimation [29,30,31,32].

Existing point cloud registration methods have achieved state-of-the-art performance on common datasets. However, most of them fail in BIM scenarios due to various challenges, noise, and data distribution from varying architectural styles, to name a few. Classic methods [33,34,35,36,37] search for hard correspondence which shows little robustness against noise since Euclidean distance is quite sensitive to offset. Feature-based methods [38,39,40,41] extract local or global reference frames in higher dimensional space to achieve registration. Most of the proposed features take not only point-level information but also line structures, greatly improving the robustness. However, feature-based methods face ambiguity during calculating the normals of boundaries on hard edges, leading to inexplicit representation. Learning-based methods [42,43,44,45,46], however, stand promising because they learn features for establishing correspondence and utilize deep neural networks to reduce the calculations. The only problem is whether they generalize well enough for different clouds.

According to 2D image theory, regions of richer geometric or semantic information can produce more discriminant knowledge. In image classification, more attention is paid to key point detection (SIFT [47], SURF [48], ORB [49]). Likewise, researchers have proposed similar works in 3D point cloud processing (USIP [50], SDK [51]). However, such key points in the cloud tend to face ambiguity during calculating their normals because of the selected neighbors located on different planes.

Motivated by the issues aforementioned, instead of using point-level information alone, this work suggests learning combined geometric knowledge of point, line, and mesh levels to alleviate the impact of ambiguous normals, and hence, improve the generalization of point cloud features. To be specific, we define a cone within the neighbors of a given centroid and calculate the three angles near its apex to form a descriptor representing its local geometry. In this case, a trade-off is made between ambiguous point normals and explicit cone angles.

Existing learning-based works evaluate their losses regarding spatial registration accuracy, while we suggest feature-matching precision of equal importance for better feature extraction in various transformations. We introduce another loss, which calculates the distances between two high-dimensional feature descriptors.

Major contributions of this article are:1Introduce a boundary-encouraging point cloud feature, PMD, to represent local geometry with higher generalization for registration, as well as solve the normal ambiguity problem.2Introduce feature matching loss to the feature extractor to produce consistent hybrid representation.3Our PMDNet shows state-of-the-art performance and higher generalization on samples from different distributions. Moreover, high performance still can be observed even when the clouds become more sparse as the distance increases.

## 2. Related Work

### 2.1. Classic Registration Methods

Among classic algorithms (Figure 1a)), ICP [33] and its variants [34,35] try to search the correspondences to minimize the distance (usually Euclidean distance) loss between the projections and the destination points or planes. However, the optimization tends to fall into local optimum because the objective function has multiple local extremums [52], especially in BIM scenarios which usually consist of millions of points. Thus, these methods usually include both coarse and fine registration, where the former aims to provide a better initialization for the latter. To enhance the robustness against noise and outliers, the following researchers view registration as a probabilistic distribution problem, where the input point clouds are treated as two distributions from the same probabilistic model [36,37]. Despite saving efforts to establish the correspondences, they still require a proper initialization because of their non-convex loss function.

### 2.2. Feature-Based Registration Methods

Instead of establishing Euclidean-based correspondences, feature-based methods extract the local reference frame (LRF) for each point in the input clouds to form feature-based correspondences. These descriptors must be distinct from each other, invariant to transformation, and robust to noise and outliers. Despite such high demands it takes, researchers proposed unique descriptors. PFH [38] calculates the invariant pose of a centroid and its neighbors in high dimensional space. FPFH [39] improves PFH with time efficiency by reducing the dimension of histograms but preserves the local feature. SHOT [40] instead focuses on normals by encoding the normal histogram in different coordinates, generated by concatenating all the local histograms. Furthermore, using Hough voting, PPF [41] calculates the 6-D pose of an LRF in a centroid-off manner. It ignores all the global coordinates and only consists of related information of normals, translation, and angles. However, mesh structures, which outmatch points and lines regarding robustness and continuity, are barely included.

### 2.3. Learning-Based Registration Methods

Early methods estimate a good initial transformation for the ICP baseline [38,53,54]. Nevertheless, recent works utilize deep neural networks to calculate a global or local reference frame for each point and then iteratively solve the transformation (Figure 1b) [55,56] firstly introduced the CNNs to the point cloud tasks, followed by numerous deep learning methods to achieve registration [42,43,44]. Among those, PointNetLk [45] uses PointNet to calculate the global features of input clouds and minimize the distance between them. DCP [46] tries to search soft correspondence with the transformer and solve the registration by SVD. These works achieved state-of-the-art performance on ModelNet and other common datasets. However, being specific to objects, they fail to generalize. On another, we notice an ambiguity (see Section 3.1) during the calculation of point normals.

### 2.4. Registration in BIM

In architecture, registration is usually performed via multiple approaches (between images, images and clouds, or clouds). Due to the huge number of points, many methods learn planar geometry instead of local reference frame from regular structures. Turning to planar geometries helps reduce the calculation and improves the overall accuracy due to declined influence of outliers. Taking plane-based registration as an example, plane structures are extracted from both input clouds via RANSAC-based methods [57,58,59], Hough transform [60], or clustering [61]. Afterward, correspondence is estimated using the extracted planes, whose accuracy largely depends on the planar segments and their normals. Ref. [62] homogenizes the as-built and as-planned models by extracting similar cross-sections and thus solves the registration problem, suggesting that the geometric shape of a room, if captured accurately, can be a distinguishing feature. Similarly, ref. [63] proposes a four-degree of freedom (DOF) registration problem and then decomposes it into two steps: (1) horizontal alignment achieved by matching the source and target ortho-projected images using the 2D line features and (2) vertical alignment achieved by making the height of the floor and ceiling in the source and target points equivalent. Ref. [64] segments the input clouds into plane pieces and clusters the parallel ones to eventually determine a transform matrix. However, considering geometries are extremely sensitive to density, their performance falls rapidly as the cloud density decreases because it is hard to extract the planar information on sparse clouds.

## 3. Feature Consistent Registration

Due to the normal ambiguity, hand-crafted features fail to calculate the explicit normal of boundaries on hard edges, which provides more information for registration than other points. We solve this problem by introducing another local reference frame, PMD, which learns point and mesh structures in the meantime. Also, previous works pay more attention to spatial registration accuracy, leaving behind feature-matching precision, which may help improve the representation in high dimensions.

We propose PMDNet to solve the issues mentioned above, including a feature extractor that generates a hybrid PMD feature, a parameter network to establish soft correspondence, and a solver to estimate transformation.

More specifically, PMDNet learns a soft correspondence between input clouds via hybrid PMD feature distances in an iterative way. The source cloud is transformed during each iteration *i* by the estimated transformation in the last iteration i−1. Then, hybrid features are extracted from both the source and reference cloud. In the meantime, PMDNet uses a parameter network to learn the annealing parameters, α,β, to refine the soft correspondence. The transformation is generated by Sinkhorn [65] algorithm. Finally, we calculate two losses in the current iteration, $L_{fe}$ and $L_{tr}$, and back-propagate them to the feature extractor and the transformation estimator. Figure 2 illustrates the pipeline of the PMDNet. We use RPMNet [66] as the backbone.

### 3.1. PMD Feature: Local Reference Frame to Encourage Boundaries


Given two sets of points, $X=\left\{x_{j}|j=1,\ldots ,J\right\}\in R^{J\times 3}$ serving as src and $Y=\left\{y_{k}|k=1,\ldots ,K\right\}\in R^{K\times 3}$ denoted as ref. The objective is to estimate a rigid transformation $\hat{G}=\hat{G}\left(\hat{R},\hat{t}\right)\in SE\left(3\right)$ with $\hat{R}\in SO\left(3\right)$ and $\hat{t}\in R^{3}$, such that:(1)$$\begin{array}{c} \hat{R},\hat{t}=\underset{R^{*}\in SO\left(3\right),t^{*}\in R^{3}}{\arg\;\min}||X\times R^{*}+t^{*}-Y{\left|\right|}^{2} \end{array}$$

Normal Ambiguity. Given a centroid point *p*, the normal of *p* is calculated within a local reference frame consisting of *p* and its neighbors $p_{1},p_{2},\ldots ,p_{n}$. As Figure 3 shows, mesh $\Omega_{1},\Omega_{2}$ with normals $n_{1},n_{2}$ intersect at *p*. Here comes the ambiguity; *p* seems to have multiple normals, depending on which mesh its neighbors are selected.

To fix normal ambiguity, we suggest refining the PPF feature with angles of intersected meshes. PPF originally are 4D point pair features describing the surface between the centroid point $x_{c}$ and each neighbor $x_{i}$ in a rotation-invariant manner:(2)$$\begin{array}{c} PPF\left(x_{c},x_{i}\right)=(∠\left(n_{c},\Delta x_{c,i}\right),∠\left(n_{i},\Delta x_{c,i}\right),∠\left(n_{c},n_{i}\right),||\Delta x_{c,i}{\left|\right|}_{2}) \end{array}$$
where $n_{c},n_{i}$ are the normals of centroid point $x_{c}$ and its neighbors $x_{i}$. Hereby we introduce another three components to PPF, that is:
(3)$$\begin{array}{cc} PMD\left(x_{c},x_{i}\right)=( & ∠\left(n_{c},\Delta x_{c,i}\right),∠\left(n_{i},\Delta x_{c,i}\right),∠\left(n_{c},n_{i}\right),||\Delta x_{c,i}{\left|\right|}_{2},∠x_{c})\\ ∠\left(x_{c}\right)=( & ∠(\Omega \left(x_{1},x_{2},x_{c}\right),\Omega \left(x_{1},x_{3},x_{c}\right)),\\ \; & ∠(\Omega \left(x_{1},x_{2},x_{c}\right),\Omega \left(x_{2},x_{3},x_{c}\right)),\\ \; & ∠(\Omega \left(x_{2},x_{3},x_{c}\right),\Omega \left(x_{1},x_{3},x_{c}\right))) \end{array}$$
where $x_{1},x_{2},x_{3}$ are the closest neighbors of $x_{c}$, and $\Omega (p_{1},p_{2},p_{3})$ denotes a mesh consisting of three points, $p_{1},p_{2},p_{3}$. Together, $x_{c},x_{1},x_{2},x_{3}$ form a triangular cone where $x_{c}$ is the apex, and $x_{1},x_{2},x_{3}$ locate on the bottom surface. The three newly added components describe the angle of the apex. PMD solves normal ambiguity because it considers not only the angle of points, but also the angle of intersected meshes.

However, the selected points $x_{1},x_{2},x_{3}$ do not always belong to an existing mesh. Thus, we suggest that the introduced angles, $∠x_{c}$ in PMD, should be considered along with $∠\left(n_{i},\Delta x_{c,i}\right),∠\left(n_{c},n_{i}\right)$ in PPF, so as to make a trade-off between them. In this case, PMD appears to be more generalizing than the PPF or other features in that the aforementioned five angles correct each other. As is proposed in RPMNet, PMD is concatenated with $x_{c}$ and $\Delta x_{i}$, forming a 13D descriptor to capture both global and local features.

### 3.2. $L_{fe}$: Feature-Aware Loss towards Feature Consistency

Deep learning methods mainly consist of a couple of modules, including a feat extractor and an SVD for computing transformation. Previous works compute the loss regarding SVD, ignoring the extractor, leading to in-explicit connection information between them. RPMNet [66] uses a Euclidean distance loss and a second loss to encourage the inliers. Ref. [67] implements Cross-entropy between estimated and ground-truth correspondence. PointNetLk [45] calculates the loss with estimated transformation $\hat{G}$ and ground-truth $G_{GT}$. In these cases, the gradient may remain still when back-propagated to the feature extractor, leading to poor updates.

RPMNet defines the total loss $L_{tr}$ as the weighted sum of the Euclidean distance $L_{el}$ between the estimated and reference clouds, and a second loss $L_{inlier}$ to encourage inliers.
(4)$$\begin{array}{c} L_{el}=\frac{1}{N}\sum_{i=1}^{N}||P\times \hat{R}+\hat{t}-P\times R_{GT}-t_{GT}{\left|\right|}^{2} \end{array}$$
(5)$$\begin{array}{c} L_{inlier}=-\frac{1}{J}\sum_{j}^{J}\sum_{k}^{K}m_{j,k}-\frac{1}{K}\sum_{k}^{K}\sum_{j}^{J}m_{j,k} \end{array}$$
(6)$$\begin{array}{c} L_{tr}=L_{el}+\lambda_{inlier}L_{inlier} \end{array}$$

However, the $L_{tr}$ is not feature-aware. Thus, we introduce another loss, $L_{fe}$ (Equation (Equation 7)), to calculate the difference between two features. $L_{fe}$ is transformation-free and only works in feature extractor:
(7)$$\begin{array}{c} L_{fe}=\frac{1}{J}\sum_{j=1}^{J}\Gamma \left(F_{src,j},G_{GT}\right)⊕\Gamma \left(F_{src,j},\hat{G}\right) \end{array}$$
where F is the extracted 13D feature, Γ(F,G) transforms the input F with a given transformation *G*. ⊕ calculates the difference between two features.

The overall loss is now the weighted sum of $L_{fe}$ and $L_{tr}$:(8)$$\begin{array}{c} L_{total}=\left(1-\lambda_{fe}\right)L_{tr}+\lambda_{fe}L_{fe} \end{array}$$

### 3.3. Annealing Parameter Network

Instead of hard correspondence, we use soft correspondence to predict transforms:(9)$$\begin{array}{c} M_{J\times K}=\left\{m_{j,k}\right\}=\left\{e^{-\beta (||F_{x_{j}}-F_{y_{k}}|{|}^{2}-\alpha )}\right\} \end{array}$$
subject to (1) $\sum_{k=1}^{K}M_{j,k}\leq 1,\forall j$; (2) $\sum_{j=1}^{J}M_{j,k}\leq 1,\forall k$; (3) $\underset{i^{*}}{\arg\;\max}M_{j_{1},i^{*}}\neq \underset{i^{*}}{\arg\;\max}M_{j_{2},i^{*}},\forall j_{1}\neq j_{2}$. Where:*F* is the hybrid feature in high dimensional space generated by the extractor.α serves as a threshold to preserve inliers and punish outliers.β is an annealing parameter to ensure convergence.

Considering α,β are usually distinct on various datasets, RPMNet uses a parameter network that takes both source and reference point clouds as input to predict α,β in an iterative manner. To be specific, the two clouds are concatenated into $P_{J+K,3}$. After that, another column in which all the elements are either 0 or 1 is added to P. Finally, P is fed to a PointNet baseline with softmax activation in the final layer to ensure the predicted parameters are always positive.

## 4. Experiments

We use ModelNet40 [68] as a common registration problem to train our PMDNet. It contains 12,311 samples of 40 categories. Figure 4 illustrates part of its samples. The clouds used for PMDNet are furthermore subsampled to 512 points for reducing the calculation. We conducted extensive experiments to evaluate the detailed performance of PMDNet against other methods.

In each experiment, we follow approximately the same process. Firstly, the raw cloud of 2048 points is input. Then, a rigid transformation matrix M∈SE(3) is generated with random rotation between $\left[0,45^{\circ }\right]$ and random translation between [−0.5,0.5] about each axis. Afterward, a copy of the raw serves as the reference cloud (*Y*), and another copy is used as the source (*X*), which will be transformed afterward by *M*. Both *X* and *Y* are furthermore shuffled and subsampled randomly to 512 points. Finally, The source (*X*) and reference (*Y*) clouds are input to PMDNet to get the estimated transformation $\hat{G}=\hat{G}\left(\hat{R},\hat{t}\right)$. which would be evaluated against $G_{GT}=G_{GT}\left(R_{GT},t_{GT}\right)=M^{-1}$ using the metrics aforementioned. Figure 5 illustrates the whole process.

The parameters of our PMDNet are listed in Table 1.

All the competing methods are evaluated using their pre-trained models on ModelNet40.

### 4.1. Metrics

All the metrics are based on the rotation and translation errors:(10)$$\begin{array}{c} Error\left(R\right)=∠\left(R_{GT}^{-1}\hat{R}\right),Error\left(t\right)=\left|\right|t_{GT}-\hat{t}{\left|\right|}^{2} \end{array}$$
where {$R_{GT},\hat{R}$} and {$t_{GT},\hat{t}$} denote the ground-truth and estimated rotation and translation, respectively. $∠\left(A\right)=\arg\;\cos\left(\frac{tr(A)-1}{2}\right)$ returns the angle of rotation matrix *A*. We provide both mean square error (MSE) and mean absolute error (MAE) for consistency with previous works [46]. All the metrics are listed in Table 2. ChamferDistance(CD) is universally used in registration problems, however, it is extremely sensitive to outliers considering $CD_{inlier}\approx 0$ and $CD_{outlier}\gg 0$. Hence, we clip this distance with a threshold of d=0.1 for mitigation. In addition, considering that cloud computing technologies are nowadays widely used in many fields but are often constrained by resources [69], time efficiency is also an essential metric to evaluate the likelihood of a model being deployed to mobile devices.
(11)$$\begin{array}{c} ERM=\frac{1}{J}\sum_{j}^{J}\left|Error{\left(R\right)}_{j}\right| \end{array}$$
(12)$$\begin{array}{c} ETM=\frac{1}{J}\sum_{j}^{J}\left|Error{\left(t\right)}_{j}\right| \end{array}$$
(13)$$\begin{array}{c} CD\left(X,Y\right)=\frac{1}{\left|X\right|}\sum_{x\in X}\min_{y\in Y}||x-{y||}^{2}+\frac{1}{\left|Y\right|}\sum_{y\in Y}\min_{x\in X}||x-{y||}^{2} \end{array}$$
(14)$$\begin{array}{c} CCD\left(X,Y\right)=\sum_{x\in X}\min(\min_{y\in Y}(||x-{y||}^{2}),d)+\sum_{y\in Y}\min(\min_{x\in X}(||x-{y||}^{2}),d) \end{array}$$

### 4.2. Ablation Experiment

In this subsection, we compare the contribution of different components of our PMD feature to determine the best choice in afterward experiments. According to its definition, we set up five controlled groups with various components to determine their influence on the overall performance of clean, noise, and unseen data.

Table 3 illustrates all the detailed results. Most groups show bare differences in noise scenarios, however, their performance on clean and unseen vary largely from one to another. $PMDNet_{A}$ achieves the best of all, reaching up to over 90% Recall, outperforming $PMDNet_{E}$ (which is the same definition with PPF) by over 10% Recall on clean and unseen. Comparing $PMDNet_{B}$ and $PMDNet_{C}$, it is easy to find that, $∠x_{c}$ alone fails to improve the representation of local geometry. $∠x_{c}$ has worsened the learning, leading to an obvious decline in accuracy. It goes the same with $∠\left(n_{i},\Delta x_{c,i}\right),∠\left(n_{c},n_{i}\right)$ despite $∠\left(n_{i},\Delta x_{c,i}\right),∠\left(n_{c},n_{i}\right)$ only lead to a slight decline on clean and unseen. Only when $∠\left(n_{i},\Delta x_{c,i}\right),∠\left(n_{c},n_{i}\right)$ and $∠x_{c}$ are both included at the same time, can they together improve the registration greatly. In previous sections, we have made a reasonable explanation for this when introducing the PMD (Section 3.1). We suggest the added angles in PMD, $∠x_{c}$, and the original ones in PPF, $∠\left(n_{i},\Delta x_{c,i}\right),∠\left(n_{c},n_{i}\right)$ be considered together, so they can compensate each other to achieve higher performance and generalization. Because $∠x_{c}$ only works for those boundaries on multiple hard edges where $∠\left(n_{i},\Delta x_{c,i}\right),∠\left(n_{c},n_{i}\right)$ fails to calculate an explicit normal. This explains the huge gap between $PMDNet_{A}$ and $PMDNet_{B},PMDNet_{C},PMDNet_{s}E$.

**Table 3 sensors-22-09694-t003:** Ablation results. $PMDNet_{A},PMDNet_{B},PMDNet_{C},PMDNet_{D},PMDNet_{E}$ is the same as introduced in Table 4. **Bold** and underline denote best and second best performance.

ID	Scene	MAE(R)↓	MAE(T)↓	CCD($1e^{-3}$)↓	Recall↑ (1.0, 0.1)	Recall↑ (0.1, 0.01)
$PMDNet_{A}$	Clean	**0.0467**	**0.00039**	**0.003226**	**99.83**%	**91.76%**
$PMDNet_{B}$	Clean	0.1680	0.00116	0.019487	98.50%	44.67%
$PMDNet_{C}$	Clean	0.0939	0.00066	0.007329	99.25%	76.70%
$PMDNet_{D}$	Clean	0.3773	0.00216	0.062771	95.59%	45.25%
$PMDNet_{E}$	Clean	0.1026	0.00075	0.006270	99.25%	76.45%
$PMDNet_{A}$	Noise	**1.1201**	**0.00990**	0.840589	80.28%	1.33%
$PMDNet_{B}$	Noise	1.1571	0.01000	**0.840061**	**81.19%**	1.16%
$PMDNet_{C}$	Noise	1.1957	0.01040	0.869413	79.36%	0.74%
$PMDNet_{D}$	Noise	1.4199	0.01190	0.998256	62.14%	0.83%
$PMDNet_{E}$	Noise	1.1365	0.01010	0.856646	80.61%	**1.49%**
$PMDNet_{A}$	Unseen	**0.0423**	**0.00037**	**0.003137**	**100.00%**	**92.02%**
$PMDNet_{B}$	Unseen	0.1642	0.00117	0.022608	98.65%	44.70%
$PMDNet_{C}$	Unseen	0.0858	0.00063	0.006717	99.68%	78.27%
$PMDNet_{D}$	Unseen	0.1947	0.00148	0.045716	97.70%	49.28%
$PMDNet_{E}$	Unseen	0.0803	0.00058	0.005054	99.52%	80.64%

In the following experiments, we keep the components in $PMDNet_{A}$ since it outperforms other groups. All the following PMDNet refers to $PMDNet_{A}$.

### 4.3. Registration Comparing on ModelNet40

#### 4.3.1. Generalization Capability

First, we provide the performance on clean data of each method in Table 5, along with the qualitative results of our method in Figure 6. In this case, all the methods are trained, tested, and evaluated on the whole ModelNet40 dataset.

We can see DeepGMR and FGR achieve the best performance on clean data, outmatching PMDNet and RPMNet by approximately 1%, which means the differences between DeepGMR, FGR, RPMNet, and our PMDNet are quite bare. Actually, except for ICP and IDAM, the remaining methods roughly reach the same accuracy and Recall, proving their performance on basic clean data.

Then, we evaluate the performance of unseen data, to test the generalization capability of each competing method. The training set only consists of the first twenty categories of ModelNet40, and all the competing methods are evaluated on the remaining twenty categories. This experiment is quite challenging for the generalization of point cloud features because all the data used in the evaluation is never seen during training. The more generalizing the feature is, the more promising results it outputs.

Table 6 shows the comparison of all the candidate methods. Our method achieves the best performance and greatly outmatches the second place, RPMNet, and other methods, reaching a 100% Recall. DeepGMR, FGR, and IDAM show a large decline (over 70% to less than 10%) here compared with their performance on clean data. Figure 7 illustrates the qualitative results of our method. Be advised that, ICP is not listed here since it is not a learning-based method.

Generally, our method achieves state-of-the-art performance both on unseen categories and clean data.

#### 4.3.2. Gaussian Noise

In this experiment, we evaluate the robustness of each competing method in presence of Gaussian noise. After subsampled, each pair of inputs, the source cloud, and the reference cloud is randomly noised with N(0,0.01) and clipped to [−0.05,0.05] to prevent extreme outliers, respectively. In this case, the one-on-one correspondence in dense clouds is corrupted due to the noise. That’s why we have been using sparse clouds from the beginning.

Table 7 illustrates the results. RPMNet and DCP v2 are the top 2 best methods in this experiment, reaching less than 1deg loss and over 90% Recall. Taking ICP (6.5 MAE(R), 0.05MAE(t), 77% Recall) as a baseline, we divide the methods into two categories, those worse than ICP (DeepGMR, FGR, and IDAM) and those better than ICP (RPMNet, DCV v2, and PMDNet). Despite being less accurate than DCP v2 or RPMNet, PMDNet still achieves acceptable results, reaching 80% Recall.

We attribute the vulnerability of PMDNet to our PMD feature. It solves the normal ambiguity of boundaries by introducing mesh angles. However, noise causes the coordinates of points to shift—on the one hand, the original boundaries are off the surrounding meshes; and on the other hand, points that are originally not boundaries become the boundaries.

### 4.4. Registration Comparing on BIM Scenarios

In this section, we test the performance of each competing method on BIM scenarios, more specifically, on clouds of uniform density and clouds with varying density, using their pre-trained models. We select 30 CAD models(.dwg) and uniformly subsample a cloud(.ply) on each of them, consisting of points varying from 0.1 million to 1.0 million. During evaluating, the ground-truth rotation $R_{GT}$ and translation $t_{GT}$ is fixed to $\left[45^{\circ },45^{\circ },45^{\circ }\right]$ and [1,1,1], respectively. Other settings are still the same as introduced above.

#### 4.4.1. Clouds of Uniform Density

In this experiment, to test the basic performance on BIM scenarios, only the dataset is changed from ModelNet40 to BIM clouds, the density of which is left unchanged.

Table 8 illustrates the results of each competing method. PMDNet outperforms any other method. Compared with others, the error of PMDNet is only about 20% of that of others, and the CCD is even one ten thousand of that of the second best. Meanwhile, It is worth noting that PMDNet has once again achieved a 100% Recall, twice as high as that of DCP v2. Figure 8 visually illustrates the input and output of PMDNet.

We perform another robust experiment to examine the robustness of candidate methods in noisy BIM scenarios. In this section, a random noise from N(0.01,0.01) is added and clipped to [–0.05, 0.05] like aforementioned.

Table 9 illustrates the results of each competing method. PMDNet fails on exactly three samples, which leads to high MAE(R) and ERM. Despite the three failed samples, the overall performance of PMDNet is still of top class since the Recall is 90%.

#### 4.4.2. Clouds with Varying Density

Our PMDNet achieves great performance on density-uniform scenarios. However, scan in real BIM practice shows a tendency to fade as the distance increases, which leads to sparse points. In this section, a declining density is applied to simulate the real scans, besides which the other settings remain the same as above. Also, we perform experiments on both clean and noisy datasets.

Density Sampling & Noise. For each cloud $P\in R^{M\times 3}$, we select a certain axis of XYZ coordinates, and apply sigmoid and normalize function to calculate a probability $\rho_{1}$, of all *M* points, which will be applied during sampling.
(15)$$\begin{array}{c} P_{\omega }\in R^{M}=\mu \times \frac{1}{1+e^{-\sigma \times P_{x}}} \end{array}$$
(16)$$\begin{array}{c} \rho_{1}\in R^{M}=\frac{P_{\omega }}{\sum P_{\omega }} \end{array}$$
(17)$$\begin{array}{c} P_{\epsilon }\in R^{M}=N\left(0,0.01\right)\times P_{\omega } \end{array}$$
where μ and σ are two introduced weights. $\rho_{1,i}$ is the probability of $P_{i}$ to be selected during the afterward sampling. $P_{\epsilon }$ is a density-aware Gaussian noise.

Figure 9 visually illustrates the input and output of PMDNet. Table 10 illustrates the metrics of each method on clean data. It is obvious that PMDNet is still outstanding, reaching MAE(R)≤0.1, MAE(t)≤0.001, 100% Recall. Furthermore, it is interesting that, compared to density-uniform Clean results (Table 8), PMDNet achieves better performance here, while others make little progress. A major factor contributing to this is the higher continuity of the PMD feature. Although the clouds become sparser and noisier with increasing distance, the nearby points remain accurate and dense, producing more information than uniform sampling, which is exactly what was applied in the density-uniform experiment.

Table 11 summaries the results on noise data. Be noticed that, PMDNet fails again on three samples. The major reason is discussed in Section 4.3.2. $∠\left(x_{c}\right)$ produces a less accurate representation since the mesh geometry is corrupted by noise. On the other hand, these three samples consist of mostly planes, with little other geometry for learning. Comparing Table 10 and Table 11, we can see a larger decline in accuracy, where MAE(R) rises from 0.0622 to 4.6186, and MAE(t) from 0.00041 to 0.03232, than that in the density-uniform experiment, where MAE(R) rises from 0.1447 to 3.7088, and MAE(t) from 0.00089 to 0.01696.

### 4.5. Time Efficiency

Time efficiency is another important metric regarding registration methods. Thus, we finally test our PMDNet against other learning-based methods in terms of time efficiency. All the methods are evaluated on Windows 11, Intel i5 12400f, NVIDIA RTX 3060TI, 16 GB 3200 MHz RAM.

Table 12 illustrates the results. We perform this experiment on clouds of 512 and 1024 resolution, with each method fixed to 3 iterations. IDAM is definitely the fastest of all methods in this study, followed by DCP v2. Despite being slower than IDAM and DCP v2, PMDNet is over 40% faster compared to RPMNet.

## 5. Conclusions

In this work, we first introduce a novel local reference frame, the PMD feature, to solve the normal ambiguity of boundaries. Moreover, as we suggest feature matching precision of equal importance as spatial accuracy, we introduce feature loss to registration.

Extensive experiments show our PMDNet achieves state-of-the-art performance. More specifically, PMDNet achieved a 100% Recall in the Unseen scenario with the generic dataset, which is 25% higher than the second-best, RPMNet. Meanwhile, PMDNet also achieved a 100% Recall on density-decreasing BIM scenarios. Last but not least, PMDNet is 40% faster than RPMNet, and 16% slower than DCP v2. However, it is 3× slower than IDAM.

PMD feature is defined to encourage the key points, however, coordinates show little robustness against noise. That’s why an obvious decline in accuracy is witnessed on noisy clouds. We are interested in further studies focused on robust detection and feature extraction on key points.

## Figures and Tables

**Figure 1 sensors-22-09694-f001:**
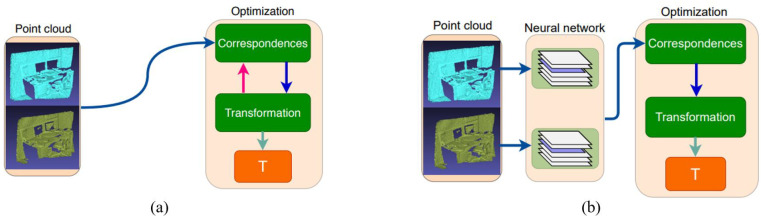
(**a**) Classic optimization-based method. (**b**) Learning-based method.

**Figure 2 sensors-22-09694-f002:**
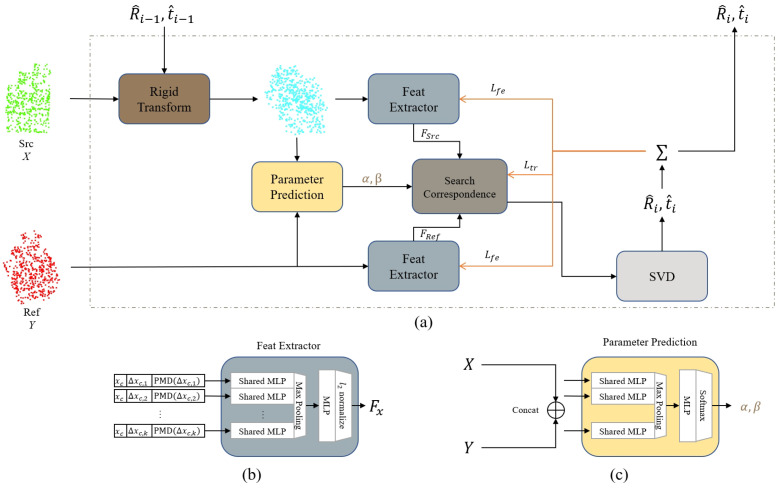
(**a**) PMDNet overview. Loss propagation is shown in orange. (**b**) Feature extractor. (**c**) Annealing parameter prediction network.

**Figure 3 sensors-22-09694-f003:**
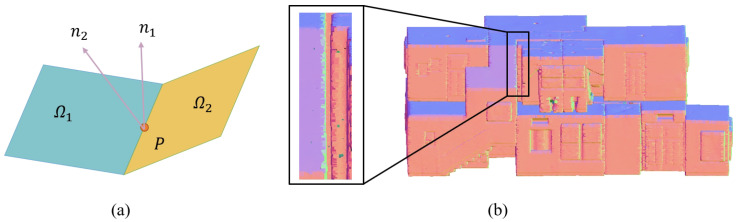
(**a**) Normal Ambiguity. The normal of boundary *p* is not explicit, instead depended on which mesh the neighbors are located during calculation. (**b**) an example of normal ambiguity. Normals are differently colored according to their orientations. The boundary between the zoomed purple and green surfaces is not explicit.

**Figure 4 sensors-22-09694-f004:**
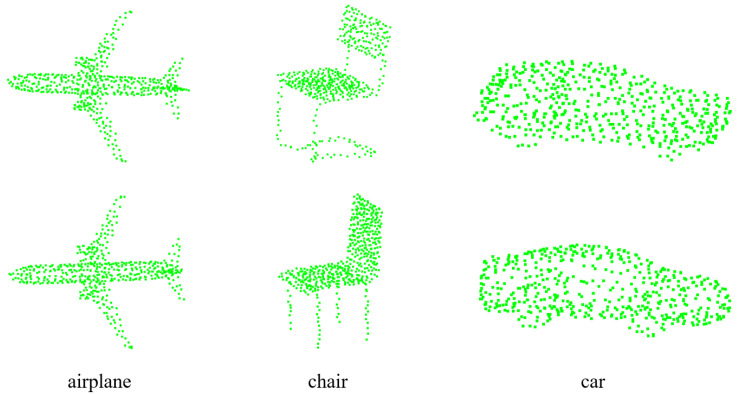
Example of ModelNet40 samples.

**Figure 5 sensors-22-09694-f005:**
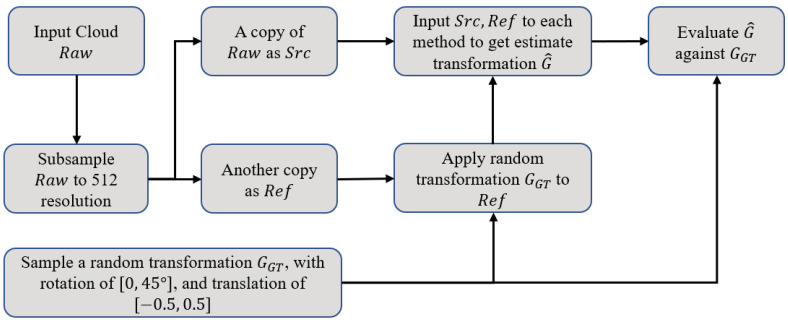
Experiment process.

**Figure 6 sensors-22-09694-f006:**
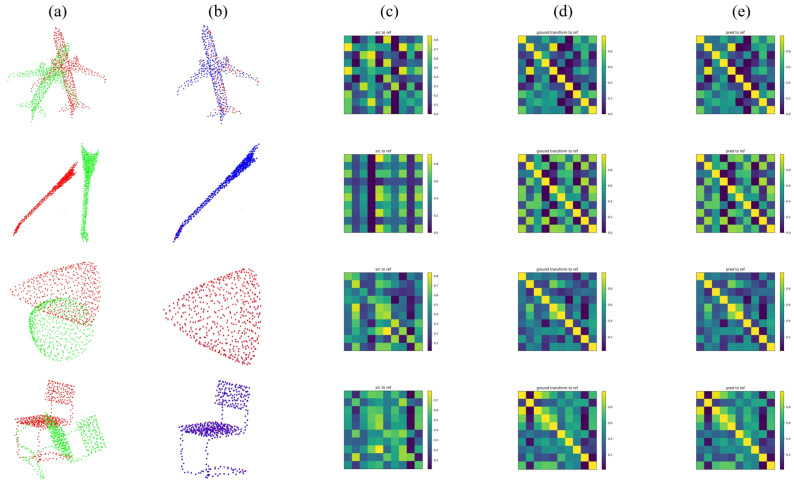
Qualitative results of the PMDNet on clean data. (**a**) source and reference clouds. (**b**) reference and predicted clouds. (**c**) correspondence between source and reference. (**d**) ground-truth correspondence. (**e**) correspondence between predicted and reference cloud.

**Figure 7 sensors-22-09694-f007:**
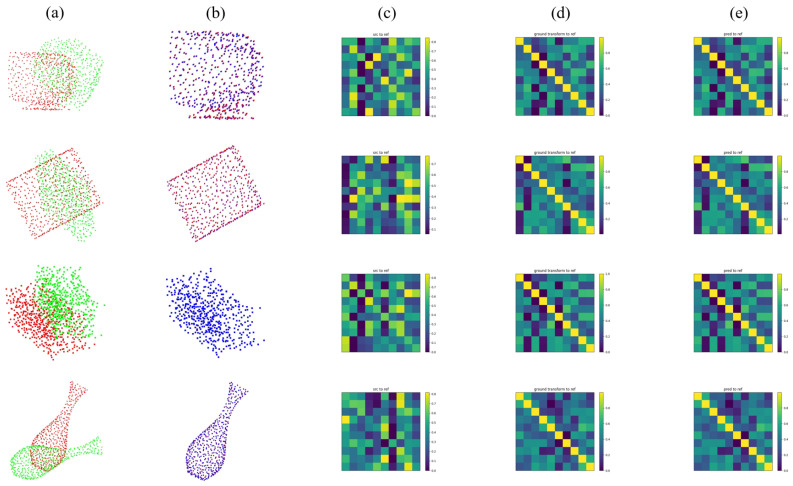
Qualitative results of the PMDNet on unseen categories. (**a**) source and reference clouds. (**b**) reference and predicted clouds. (**c**) correspondence between source and reference. (**d**) ground-truth correspondence. (**e**) correspondence between predicted and reference cloud.

**Figure 8 sensors-22-09694-f008:**
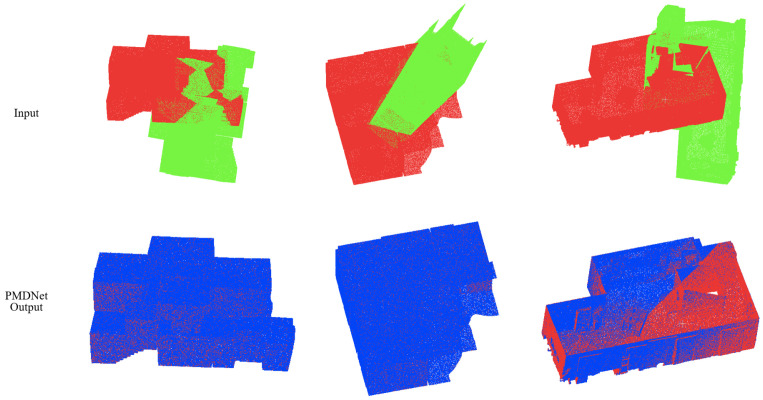
Qualitative results of the PMDNet on density-uniform clean BIM data. src, ref, and pred clouds are colored green, red, and blue, respectively.

**Figure 9 sensors-22-09694-f009:**
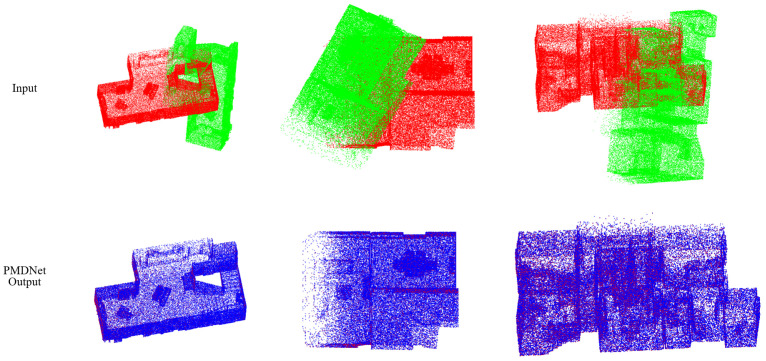
Qualitative results of the PMDNet on density-decreasing clean BIM data. src, ref, and pred clouds are colored green, red, and blue, respectively.

**Table 1 sensors-22-09694-t001:** Parameters of PMDNet.

Parameter	Value
learning rate	$1\times 10^{-4}$
epochs	1024
batch size	8
optimizer	Adam

**Table 2 sensors-22-09694-t002:** Evaluate Metrics.

Metrics	Ref Equation	Notes
err_r_deg_mean (ERM)	Equation (Equation 11)	Mean of isotropic Error of Rotation
err_t_mean (ETM)	Equation (Equation 12)	Mean of isotropic Error of Translation
CCD	Equation (Equation 14)	Clip Chamfer Distance
MAE(R)	-	Mean Absolute Error of Rotation, in the unit of degrees
MAE(T)	-	Mean Absolute Error of Translation, in the unit of degrees
Recall(ω,δ)	-	Proportion of samples with, MAE(R) < ω(${}^{\circ }$) and MAE(t) < δ(m)

**Table 4 sensors-22-09694-t004:** Ablation setup. $PMDNet_{A},PMDNet_{B},PMDNet_{C},PMDNet_{D},PMDNet_{E}$ select different components of the PMD feature.

ID	$x_{c}$	$x_{c}-x_{i}$	$∠(x_{c})$	$∠(n_{r},n_{i})$
$PMDNet_{A}$	✓	✓	✓	✓
$PMDNet_{B}$	✓	✓	✓	
$PMDNet_{C}$	✓	✓		
$PMDNet_{D}$	✓			✓
$PMDNet_{E}$	✓	✓		✓

**Table 5 sensors-22-09694-t005:** Results on clean data. **Bold** and underline denote best and second best performance.

Method	MAE(R)↓	MAE(t)↓	ERM↓	ETM↓	CCD↓	Recall(1.0, 0.1)↑
ICP	6.4467	0.05446	3.079	0.02442	0.030090	74.19%
FGR	**0.0099**	0.00010	0.006	0.00005	0.000190	99.96%
RPMNet	0.2464	0.00050	0.109	0.00050	0.000890	98.14%
IDAM	1.3536	0.02605	0.731	0.01244	0.044700	75.81%
DeepGMR	0.0156	**0.00002**	**0.001**	**0.00001**	0.000030	**100.00%**
PMDNet	0.0467	0.00039	0.087	0.00081	**0.000003**	99.83%

**Table 6 sensors-22-09694-t006:** Results on unseen data. **Bold** and underline denote best and second best performance. They are trained on the first twenty categories and tested on the remaining categories.

Method	MAE(R)↓	MAE(t)↓	ERM↓	ETM↓	CCD↓	Recall(1.0, 0.1)↑
FGR	41.9631	0.29106	23.950	0.14067	0.123700	5.13%
RPMNet	1.9826	0.02276	1.041	0.01067	0.087040	75.59%
IDAM	19.3249	0.20729	10.158	0.10063	0.129210	0.95%
DeepGMR	71.0677	0.44632	44.363	0.22039	0.147280	0.24%
DCP v2	2.0072	0.00370	3.150	0.00503	NA	NA
PMDNet	**0.0423**	**0.00037**	**0.076**	**0.00075**	**0.000003**	**100.00%**

**Table 7 sensors-22-09694-t007:** Results on Gaussian noise data. **Bold** and underline denote best and second best performance.

Method	MAE(R)↓	MAE(t)↓	ERM↓	ETM↓	CCD↓	Recall(1.0, 0.1)↑
ICP	6.5030	0.04944	3.127	0.0225	0.05387	77.59%
FGR	10.0079	0.07080	5.405	0.0338	0.06918	30.75%
RPMNet	**0.5773**	0.00532	**0.305**	0.0025	0.04257	**96.68%**
IDAM	3.4916	0.02915	1.818	0.0141	0.05436	49.59%
DeepGMR	2.2736	0.01498	1.178	0.0071	0.05029	56.52%
DCP v2	0.7374	**0.00105**	1.081	**0.0015**	NA	NA
PMDNet	1.1201	0.00990	2.224	0.0208	**0.00084**	80.28%

**Table 8 sensors-22-09694-t008:** Results on clean density-uniform BIM scenarios. **Bold** and underline denote best and second best performance, respectively.

Methods	MAE(R)↓	MAE(T)↓	ERM↓	ETM↓	CCD↓	Recall(1.0, 0.1)↑
DCP v1	3.9788	0.00433	5.641	0.08823	0.089453	13.33%
DCP v2	1.0328	0.01319	1.415	0.02614	0.088061	50.00%
IDAM	23.7044	0.08125	50.176	0.16067	0.094456	0.00%
PMDNet	**0.1447**	**0.00089**	**0.522**	**0.00181**	**0.000009**	**100.00%**

**Table 9 sensors-22-09694-t009:** Results on noise density-uniform BIM scenarios. **Bold** and underline denote best and second best performance, respectively. PMDNet fails on three scenarios, contributing to large MAE(R) and ERM, but achieves great performance on all the other samples. PMDNet${}^{†}$ shows the metrics on successfully predicted scenarios.

Methods	MAE(R)↓	MAE(T)↓	ERM↓	ETM↓	CCD↓	Recall(1.0, 0.1)↑
DCP v1	3.8952	0.04443	5.528	0.08900	0.088047	10.00%
DCP v2	**0.0298**	0.02985	**1.241**	0.05893	0.088041	56.67%
IDAM	21.9374	0.08308	48.064	0.16032	0.088634	0.00%
PMDNet	3.7088	**0.01696**	10.097	**0.04232**	**0.002669**	**90.00%**
PMDNet${}^{†}$	0.3039	0.00241	0.496	0.00490	0.000442	100.00%

**Table 10 sensors-22-09694-t010:** Results on clean density-decreasing BIM scenarios. **Bold** and underline denote best and second best performance, respectively.

Methods	MAE(R)↓	MAE(T)↓	ERM↓	ETM↓	CCD↓	Recall(1.0, 0.1)↑
DCP v1	3.6161	0.95663	5.130	1.12523	0.198468	0.00%
DCP v2	0.9948	0.96918	1.410	1.11823	0.198336	0.00%
IDAM	23.0635	1.56710	26.303	1.78247	0.199977	0.00%
PMDNet	**0.0622**	**0.00041**	**0.101**	**0.00061**	**0.000002**	**100.00%**

**Table 11 sensors-22-09694-t011:** Results on noise density-decreasing BIM scenarios. **Bold** and underline denote best and second best performance, respectively. PMDNet fails on three scenarios, contributing to large MAE(R) and ERM, but achieves great performance on all the other samples. PMDNet${}^{†}$ shows the metrics on successfully predicted scenarios.

Methods	MAE(R)↓	MAE(T)↓	ERM↓	ETM↓	CCD↓	Recall(1.0, 0.1)↑
DCP v1	3.1935	0.93363	4.281	1.09541	0.198179	0.00%
DCP v2	**1.0033**	0.94744	**1.295**	1.08987	0.198062	0.00%
IDAM	23.0495	1.52381	26.616	1.71634	0.200000	0.00%
PMDNet	4.6186	**0.03232**	23.987	**0.13219**	**0.001516**	**90.00%**
PMDNet${}^{†}$	0.4358	0.00305	0.736	0.00604	0.000493	96.67%

**Table 12 sensors-22-09694-t012:** Results on time efficiency. **Bold** and underline denote best and second best performance, respectively. All the results are in the unit of milliseconds.

Points	DCP v2(3 Iters)	RPMNet(3 Iters)	IDAM(3 Iters)	PMDNet(3 Iters)
512	15.04	32.47	**5.84**	17.72
1024	18.81	38.35	**7.11**	21.58

## Data Availability

Not applicable.

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
