# Peer review of "Feature Consistent Point Cloud Registration in Building Information Modeling"

_sensors, 2022, doi:10.3390/s22249694_

Round 1

Reviewer 1 Report

In general, it is a very interesting study to deal with registration problems in BIM scenarios based on the PyramidFeature (PMD). I found the paper to be overall well described. However, there are some parts that need further clarity, as follows:

-          Page 10 – Subsection 4.4 . . subsample a cloud. .

How many percentages do the authors set for the dataset subsampling? Why?

-          Page 12 - Caption of Table 10 . . .PMDNet fails on two scenarios. . .

The authors need to explain the reasons for this.

-          I am missing the discussion part of the study

-          Page 12 – Conclusion

The authors need to elaborate further on the limitation of the proposed method and further recommendations.

Author Response

Dear Editor and Associate Editor,

Thank you very much for organizing the review process. We appreciate the insightful and invaluable comments from all the reviewers and have revised the manuscript accordingly to address their concerns. In this revision, we made the following major changes (highlighted in blue throughout the manuscript) to address the concerns raised by the reviewers.

  1. We renamed the title of this work, the new title is “Feature Consistent Point Cloud Registration in Building Information Modeling”.
  2. In Section 1, we emphasized the scope, motivation, and idea of this work according to the comments from the reviewers. (Page 1, Page 2)
  3. In Section 2, we added figures for better understanding of different categories of methods for registration. (Page 2)
  4. In Section 3, we
    • renamed the section title as “Feature Consistent Registration” to give a brief idea of our method. (Page 4)
    • clarified our PMD feature description. (Page 5)
    • introduced three modules of our network, and discuss them in detail in the following three subsections. (Page 4, Page 5, Page 6)
  5. In Section 4, we
    • gave an example of ModelNet40 samples. (Page 6)
    • clarified our experiment setup with description and process figure, along with parameter configuration. (Page 7)
    • cited the paper the reviewer recommended in metric description. (Page 7)
    • explained why subsampling is applied to CAD models. (Page 11)
    • elaborated “Cloud with Consistent Density”, and rename it to “Cloud of Uniform Density”. (Page 11)
    • analyzed the reason why PMDNet fails on certain samples with noise. (Page 13)
    • added analysis of time efficiency experiment. (Page 13)
  6. In Section 5, we
    • emphasized the contribution of this work. (Page 13)
    • gave experiment summaries. (Page 14)
    • summarized the limitations of PMDNet, as well as provided suggestions for future works. (Page 14)

Attached please find the detailed point-to-point responses to all comments, together with the revised manuscript, for your further consideration.

Reviewer 2 Report

1) The definition of PMDnet is unclear. It is not appropriate as a title.

2) The explanation is not faithful in the introduction. The author needs to clearly present the exact method and scope of research.

3) In 2. related work, the author needs to explain related information in a table or figure so that the reader can clearly check it.

4) In 3. method, Chapter 3 is not clear in the title and content. The proposed PMD is unclear. First of all, the author needs to separate the contents of PMD into separate chapters and explain in detail, and present the development concept in detail.

5) In 4. experiments, the author needs to clearly present 12,311 samples and 40 categories, and further explain the sample case as an example.

6) In 4. experiments, the author needs to present the step-by-step process and process of the experiment in detail in figures.

7) In the conclusion, the author described the meaning of technology development. It has not been described at all in what aspects of academic contribution to the conclusion. Although this paper is recognized as a technical article, it lacks a lot of academic value.

Author Response

(The authors gave the same response as above.)

Reviewer 3 Report

- Paper discusses the Point Cloud Registration in building information modeling (BIM). This is interesting to the readers. Appreciated.

- On Page 02, after 3rd Paragraph, Line "Generally, this work is interesting in the following aspects:"......in place of this sentence, use "Major Contributions" of this article are:"

- It seems that the authors missed recent relevant experimentation setup, So it is difficult to recreate the experiments the authors have made. Improve it.

- Elaborate "Clouds with Consistent Density" ??

- In Sub-Section 4.4, why experimentation made on 30 CAD models and sub-sample a cloud on each of them ??

- It would be appreciated if experimentation analyze on the proposed algorithm in terms of time consumed and compare with other algorithms. (Ref. Subsection 4.5 improve).

- Conclusion can be improved on outcome of this experimentation conduct aspect before & after.

-Cite below article to improve the readability of this paper:

(a) Performance Evaluation of an Adopted Model based on Big-Bang Big-Crunch and Artificial Neural Networks for Cloud Applications", Pradeep Singh Rawat, Robin Singh Bhadoria, Puneet Gupta and G. P. Saroha, Kuwait Journal of Science, Vol. 48, No. 04, pp. 1-13, Oct 2021.

Author Response

(The authors gave the same response as above.)

Round 2

Reviewer 1 Report

Dear Authors,

Thanks for addressing my feedback.

I satisfy with the current manuscript.